**Data Availability Statement:** All relevant data are within the paper and its Supporting information files.

# The reliability of the angle of deviation measurement from the Photo-Hirschberg tests and Krimsky tests

**S. Tengtrisorn***, **A. Tungsattayathitthan, S. Na Phatthalung, P. Singha, N. Rattanalert, S. Bhurachokviwat, S. Chouyjan**

Department of Ophthalmology, Faculty of Medicine, Prince of Songkla University, Hat Yai, Songkhla, Thailand

* tsupapor@medicine.psu.ac.th

## Abstract

### Objective

To compare the angle of deviation measured from Photo-Hirschberg testing and Krimsky testing, with that from an alternate prism cover test (APCT) in strabismus patients.

### Methods

A cross-sectional study was conducted in Songklanagarind Hospital, Thailand. Thirty-three strabismus patients were photographed for analysis by Photo-Hirschberg testing using computer software. The corneal light reflex displacement, converted into prism diopter (PD), was compared to the angle of deviation measured with APCT. Twenty-eight strabismus patients were tested with the Krimsky test. Data were analyzed using Pearson correlation and paired t-tests. The study excluded 4 intermittent exotropia cases, 1 intermittent esotropia case and 2 which cases missing data for krimsky test.

### Results

The mean±SD of the deviation angle, measured by APCT with a fixation target at 30 cm and 6 m; were 48.09±16.34PD and 47.82±15.73 PD, respectively. At 1 m, the difference in the angle of deviation measured from APCT and the Photo-Hirschberg test within 10 PD were 58.8% and 63.6%, for ET and XT, respectively. The difference in the angle of deviation measured from APCT and Krimsky tests within 10 PD in ET and XT were 86.7% and 80.0%, respectively. At 4 m, the difference in angle of deviation measured from APCT and Photo-Hirschberg tests within 10 PD in ET and XT were 58.8% and 54.5%, respectively; whereas, the difference in the angle of deviation measured from APCT and Krimsky tests within 10 PD in ET and XT were 80.0% and 70.0%, respectively.

### Conclusion

The reliability of Krimsky test was better than Photo-Hirschberg test for measuring an angle of deviation.

**Funding:** This research was partially supported by a grant from the Faculty of Medicine, Prince of Songkla University. There was no additional external funding received for this study.

**Competing interests:** On behalf of all authors the corresponding author declares that no competing interests exist.

## Introduction

Currently, there are various methods for diagnosis of strabismus and measuring the angle of deviation. To date, the alternate prism cover test (APCT) is the gold standard method. However, this method requires professionalism, is time consuming, depends on a special device) prism(, and requires the cooperation of patients in the measurement. In addition, the accuracy of strabismus measurement is often limited; especially when performed by non-experts [1]. Accurate angle measurement is very important for preoperative evaluations, operative procedures and postoperative follow up. Hospitals in developing countries such as Thailand do not have enough specialists or the necessary diagnostic equipment, especially in rural areas. Thus, most eye muscle surgeries are performed in urban hospitals. Additionally, the heavy workload of ophthalmologists hinders them from strabismus examination by APCT. Therefore, cost effective and accurate alternative methods involving simple procedures that can be performed in hospitals within rural areas are required. This study looks at a simple method that uses affordable equipment to reduce that workload. In addition, this method could help reduce the travel burden for patients in rural settings for access to a tertiary care hospital. This study compares the angle of deviation measured from the Photo-Hirschberg test and Krimsky test with that from APCT. It is expected that the results will provide accurate measurement of the angle of deviation; particularly in cases of uncooperative patients.

## Methods

The research was approved by the Human Research Ethics Committee of the Faculty of Medicine, Prince of Songkla University. The study designs were approved as 2 studies: the Photo-Hirschberg test compared to APCT and the Krimsky test compared to APCT. The consent was written by subjects and parents for children younger than 18 years.

Subjects were excluded if any of the following were present: cyclovertical strabismus, amblyopia, accommodative ET, paralytic strabismus, previous strabismus surgery and abnormality of the ocular surface; such as, corneal scar or pterygium, which may disturb the CLR and confound interpretation of the study results. This study enrolled 33, horizontal strabismus subjects; 14 males (42.4%) and 19 females, at Songklanagarind Hospital; from 2014 to 2015. The mean age was 16.42±15.64 yr. The pattern of strabismus revealed 18 ET and 11 XT and 1 intermittent ET and 4 intermittent XT. Informed consent was obtained from all subjects who were included in the study, or from their guardians. The subjects received a complete eye examination and measurement of the angle of deviation with an APCT, and Krimsky test by 2 experienced orthoptists. The assistance researcher took 2 photographs with the flash on to perform the Photo-Hirschberg test, which is described later. Times to perform the test in each subject were 30 minutes, 20 minutes, 10 minutes and 20 minutes for eye examination, APCT, photography and Krimsky test, respectively.

The Photo-Hirschberg test.

Photography was performed using a Canon EOS 650D, digital camera, each subject was recorded 8 times, with a photographic resolution of 5,184 x 3,456 pixel images or 18 megapixels. The camera was placed on a stand 1 meter (m) from the subject at the level of the subject's eye with the flash on. The subject with a ruler attached to the forehead fixated on a target at 1m for near (N1) and 4 m for distance (D1). At each distance, 4 photographs were taken: occluding the left eye (LE), occluding the right eye (RE), both eyes open with RE fixation and both eyes open with LE fixation, respectively (Fig 1).

The quality of images was checked. The digital images were then transferred to a computer, and the position of the CLR, in relation to the nasal limbus, was recorded using an automated program. Fig 1, **A** is the distance from the CLR to nasal corneal limbus on the right eye while

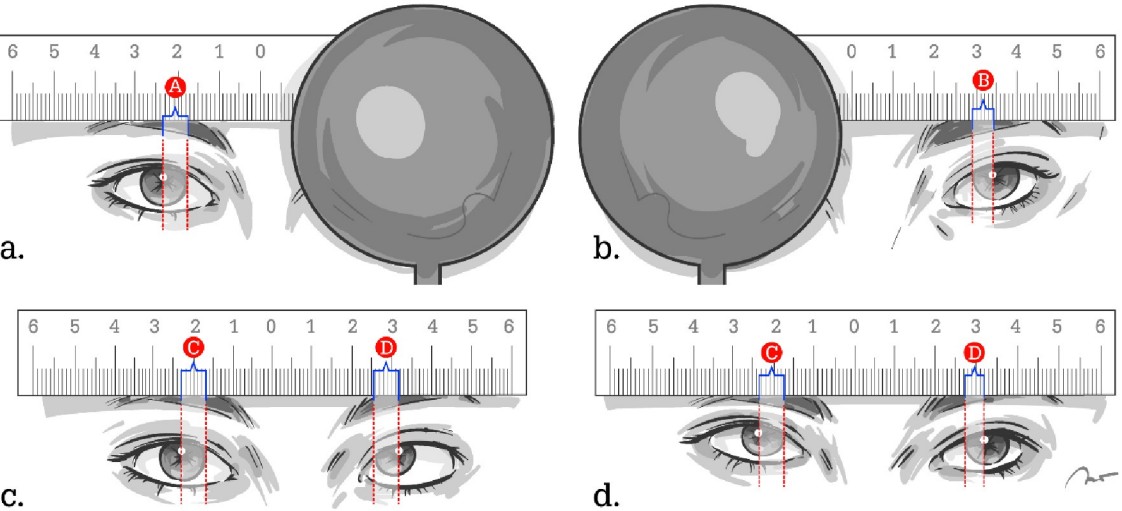

**Fig 1. The photographic method.** (a) Right eye fixing to a central target, and left eye occluded, A is the distance from the CLR to nasal corneal limbus. (b) Left eye fixing to a central target, and right eye occluded, B is the distance from the CLR to nasal corneal limbus. (c) Both eyes open, and right eye fixating on a central target, C is the distance from the right CLR to nasal corneal limbus. The distance from the left CLR to nasal corneal limbus is D. (d) Both eyes open, and left eye fixating on a central target. (The above illustrations are the original works of the authors).

the right eye is fixing, and **B** is the distance from the CLR to nasal corneal limbus on the left eye while the left eye is fixing. The sum of A and B was considered to represent virtual orthophoria. In the images with both eyes open, **C** is the distance from the CLR to nasal corneal limbus on the right eye, and **D** is the distance from the CLR to nasal corneal limbus on the left eye. The sum of C and D was considered to represent strabismus. The value of [(A + B)–(C + D)] was considered to reflect the magnitude of the angle of horizontal displacement, measured in mm, with negative values representing ET, and positive values representing XT. The magnitude of deviation was related to PD by the Hirschberg coefficients from a previous report [2]. At 1 m (N1), the CLR displacement in mm was converted into PD using a Hirschberg coefficient of 21.31 PD/mm and 17.27 PD/mm for the ET and XT values, respectively. At 4 m (D1) the conversion factors were 20.16 PD/mm and 15.63 PD/mm for the ET and XT values, respectively. The Photo-Hirschberg tests were compared with the angle of deviation obtained from APCT measurements.

The Krimsky test was performed by measuring the angles of deviation when the patient fixated on a target at 30 cm (N2) and 6 m (D2). The angles of deviation were compared with the angle of deviation obtained from APCT measurements. The fixation target at near and distance were standard for the Krimsky test and APCT.

## Sample size calculation formula for correlation

$$N = [Z_{\alpha/2} + Z_{\beta}/C(r)]^2 + 3$$

When $Z_{\alpha/2}$ = Probability of type I error

$$C(r) = Log_e\{(1 + r)/(1 - r)\}$$

$$\alpha = 0.05 \quad \beta = 0.2 \quad r = 0.4$$

The total sample size required for this study was 53.

## Statistical analyses

The relation between the angles of deviation measured by the Photo-Hirschberg test and the APCT, and that between the angle of deviation measured by the Krimsky test and APCT, were evaluated using two-way scatter plots and Pearson' Correlation (r) to describe the direction, form and strength of the relationship. The strength of linear relationship was interpreted from poor to very strong [3]. The zone between the dotted lines in Figs 3–6 represents range within ±10 PD of the APCT values. Acceptable and unacceptable areas were designated a, b, c, and d. The acceptable areas were **b**; representing the angle of deviation derived from the Photo-Hirschberg test or Krimsky test, as more than 10 PD higher than that from APCT, and **c**, representing the angle of deviation derived from the Photo-Hirschberg test or Krimsky test lower than that from APCT of less than 10 PD. The unacceptable areas were **a**, representing the angle of deviation derived from the Photo-Hirschberg test or Krimsky test of more than 10 PD higher than that from APCT; whereas, area **d**, the angle of deviation derived from the Photo-Hirschberg test or Krimsky test of more than 10 PD lower than that from APCT.

## Results

The angle of deviation from the Photo-Hirschberg test and APCT were measured from 33 strabismus subjects. The 1 intermittent ET case and 4 intermittent XT cases were excluded before analysis. The Krimsky test was performed on 31 subjects; 2 subjects having variable angles were omitted from analysis (Fig 2).

Using APCT, The ET group showed mean angle deviations of 47.06±13.09 and 46.72±12.42 at N and D, respectively. The XT group showed mean angle deviations of 49.33±19.98 and 49.13±19.36 at N and D, respectively.

With the Photo-Hirschberg test, the ET group showed mean angle deviations of 50.87 ±19.58 and 48.71±16.10 at N1 and D1, respectively; whereas, the XT group showed mean angle deviations of 33.15±18.68 and 38.20±21.56 at N1 and at D1, respectively.

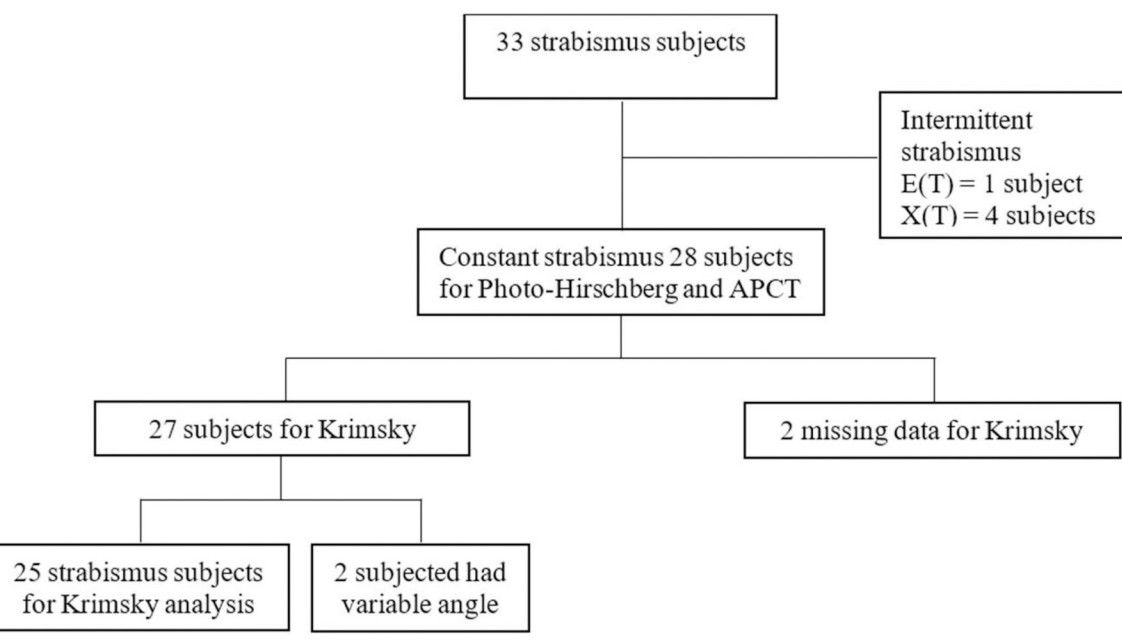

**Fig 2. Flow diagram of strabismus subjects.**

The ET group for the Krimsky test, showed mean angle deviations of 46.67±12.05 and 45.33±13.02 at N2 and at D2, respectively, and the XT group showed mean angle deviations of 50.36±14.07 and 51.43±12.32 at N2 and at D2, respectively.

## Correlation analysis

At N, the correlation coefficients(r) of the angle of deviation measured by the Photo-Hirschberg test and APCT, for ET and XT, were 0.443 and 0.637, respectively. This indicated a fair to moderately strong correlation (Fig 3). The number of subjects with differences in angle of deviation within 10 PD were 17 (60.7%), 10 (58.8%) and 7 (63.6%), for total, ET and XT, respectively (Table 1). The correlation coefficients(r) of the angle of deviation measured by the Krimsky test and APCT, for ET and XT, were 0.717 and 0.812, respectively; indicating a moderately strong correlation (Fig 4). The number of subjects with differences in angle of deviation within 10 PD were 21 (84.0%), 13 (86.7%) and 8 (80.0%), for total, ET and XT, respectively (Table 2).

At D, the correlation coefficients(r) for angle of deviation measured by the Photo-Hirschberg test and APCT were 0.501 and 0.778, for ET and XT, respectively; indicating a moderately strong correlation (Fig 5). The number of subjects with difference in angle of deviation within 10 PD were 16 (57.1%), 10 (58.8%) and 6 (54.5%), for total, ET and XT, respectively (Table 1). The correlation coefficients(r) of the angle of deviation measured by the Krimsky test and APCT were 0.696 and 0.798, for ET and XT, respectively; indicating a moderately strong correlation (Fig 6). The number of subjects with differences in angle of deviation within 10 PD were 19 (76.0%), 12 (80.0%) and 7 (70.0%), for total, ET and XT, respectively (Table 2).

This study excluded 1 intermittent ET, 4 intermittent XT patients, with angles of deviation of 27.7, 0.17, 5.01, 16.2 and 25.2 PD, determined from the Photo-Hirschberg test and angles of deviation of 20, 40, 40, 35 and 40 PD measured using APCT.

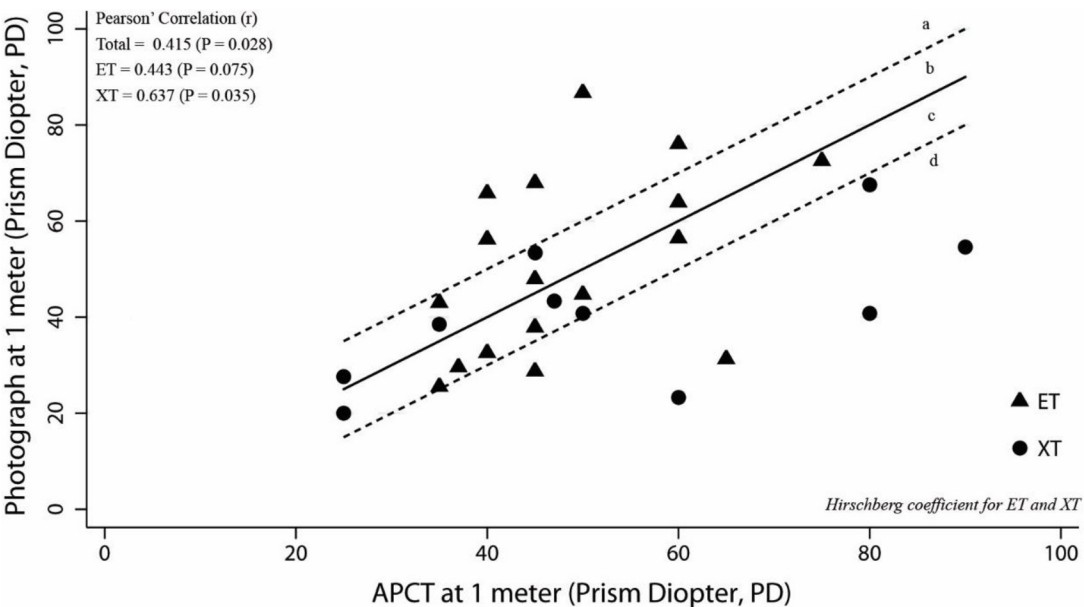

**Fig 3. The relationship between angle of deviation from the Photo-Hirschberg test with APCT at N.** The solid line is the line of equality and the zone between dotted lines represents range ±10 PD of the APCT values. The acceptable area were b and c. The unacceptable area were a and d. (ET = esotropia;XT = exotropia).

**Table 1. Summary results of correlation between angle of deviation from the Photo-Hirschberg test with the APCT.**

| Variable | At N, n(%) | | | At D, n(%) | | |
|---|---|---|---|---|---|---|
| | Total | ET | XT | Total | ET | XT |
| Absolute difference within acceptable range | 17 (60.7) | 10 (58.8) | 7 (63.6) | 16 (57.1) | 10 (58.8) | 6 (54.5) |
| Area b | 6 (21.4) | 3 (17.6) | 3 (27.3) | 8 (28.6) | 4 (23.5) | 4 (36.4) |
| Area c | 11 (39.3) | 7 (41.2) | 4 (36.4) | 8 (28.6) | 6 (35.3) | 2 (18.2) |
| Absolute difference within unacceptable range | 11 (39.3) | 7 (41.2) | 4 (36.4) | 12 (42.9) | 7 (41.2) | 5 (45.5) |
| Area a | 5 (17.9) | 5 (29.4) | 0 (0) | 4 (14.3) | 4 (25.5) | 0 (0) |
| Area d | 6 (21.4) | 2 (11.8) | 4 (36.4) | 8 (28.6) | 3 (17.6) | 5 (45.5) |

APCT = alternate prism cover test, ET = Esotropia, XT = Exotropia, PD = prism diopter, = n number of subjects, N = near fixation, D = distance fixation.

## Discussion

The accuracy of the angle of deviation depends on the measurement method, experience of the personnel as well as cooperation of the patient. The Hirschberg's test, which uses the position of corneal light reflex (CLR) related to pupillary or corneal limbus for estimation of angle of deviation, is a simple and easy test. However, the test precision is poor, because it uses the measurement of position of CLR, which is prone to human error; especially when performed by inexperienced personnel [1, 4]. The Hirschberg coefficients varies between 13 and 21 prism diopter (PD) per mm, due to variations of research methodology, techniques and subjects [5–12]. Recently, investigators [2] have used digital cameras and computers to detect the position of CLR, so as to improve the accuracy of the Hirschberg's test. This present study aimed to use conversion factors from our previous study. We defined this method as: a Photo-Hirschberg test. The past, a method using CLR combined with photographs, videos, and computer software was reported that could possibly improve diagnosis and measurement of angle of deviation [13, 14].

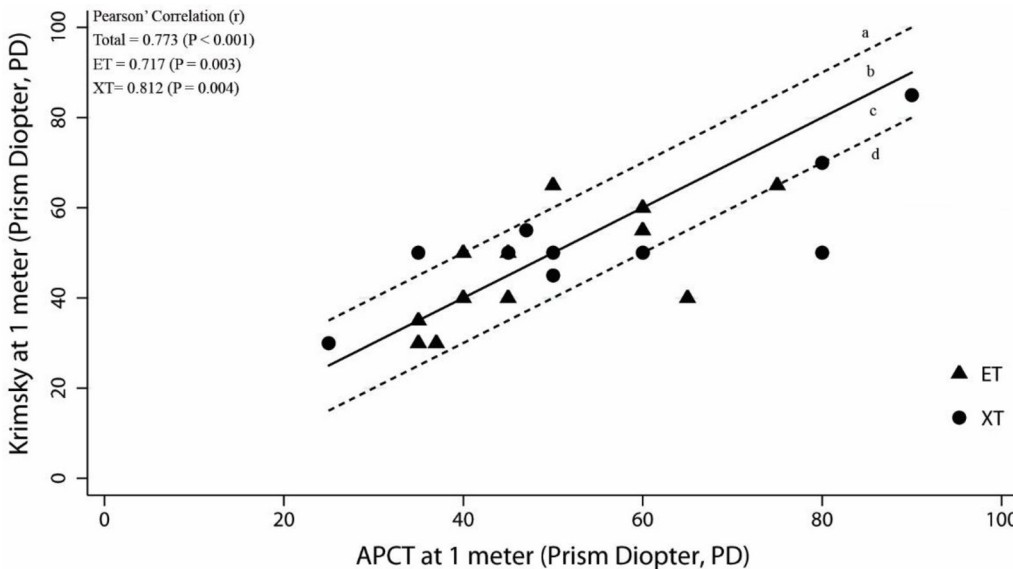

**Fig 4. The relationship between angle of deviation from the Krimsky test with APCT at N.** The solid line is the line of equality and the zone between dotted lines represents range ±10 PD of the APCT values. The acceptable area were b and c. The unacceptable area were a and d. (ET = esotropia; XT = exotropia).

**Table 2. Summary results of correlation between angle of deviation from the Krimsky with the APCT.**

| Variable | At N2, n(%) | | | At D2, n(%) | | |
|---|---|---|---|---|---|---|
| | Total | ET | XT | Total | ET | XT |
| Equality | 5 (20) | 4 (26.7) | 1 (10) | 5 (20) | 4 (26.7) | 1 (10) |
| Absolute difference within acceptable range | 16 (64) | 9 (60) | 7 (70) | 14 (56) | 8 (53.3) | 6 (60) |
| Area b | 5 (20) | 2 (13.3) | 3 (30) | 5 (20) | 2 (13.3) | 3 (30) |
| Area c | 11 (44) | 7 (46.7) | 4 (40) | 9 (36) | 6 (40) | 3 (30) |
| Absolute difference within unacceptable range | 4 (16) | 2 (13.3) | 2 (20) | 6 (24) | 3 (20) | 3 (30) |
| Area a | 2 (8) | 1 (6.7) | 1 (10) | 2 (8) | 1 (6.7) | 1 (10) |
| Area d | 2 (8) | 1 (6.7) | 1 (10) | 4 (16) | 2 (13.3) | 2 (20) |

APCT = alternate prism cover test, ET = Esotropia, XT = Exotropia, PD = prism diopter, n = number of subjects, N2 = near fixation at 30 cm, D2 = distance fixation at 6 m.

The Krimsky test, employed for measuring the angle of deviation using deviated light from a prism and the position of CLR, is simple, but depends on experienced personnel and a cooperative patient. Choi [15] observed accuracy within 10 PD in angle of deviation measurement using Hirschberg and Krimsky tests by experienced strabismologists. Holmes [16] observed that the 95% limits of agreement for inter observer for APCT at near (N) and distance (D) were 9.2 PD and 10.2 PD, respectively. Therefore, we defined the acceptable range of difference in angle of deviation between methods as within±10 PD.

Basmak [17] showed the angle Kappa was higher in XT patients than in ET patients, resulting in a biased angle measurement when performed by the Hirschberg test or Krimsky tests. The angle Kappa is the angle between the optical axis and the visual axis of the eye, which affects the position of the CLR in non-strabismus people.

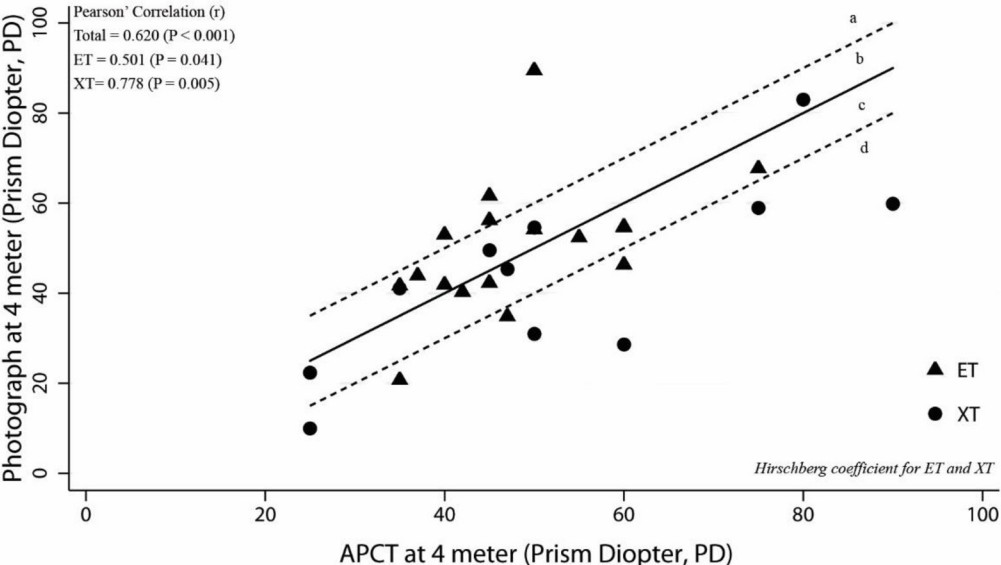

**Fig 5. The relationship between angle of deviation from the Photo-Hirschberg test with APCT at D.** The solid line is the line of equality and the zone between dotted lines represents range ±10 PD of the APCT values. The acceptable area were b and c. The unacceptable area were a and d. (ET = esotropia; XT = exotropia).

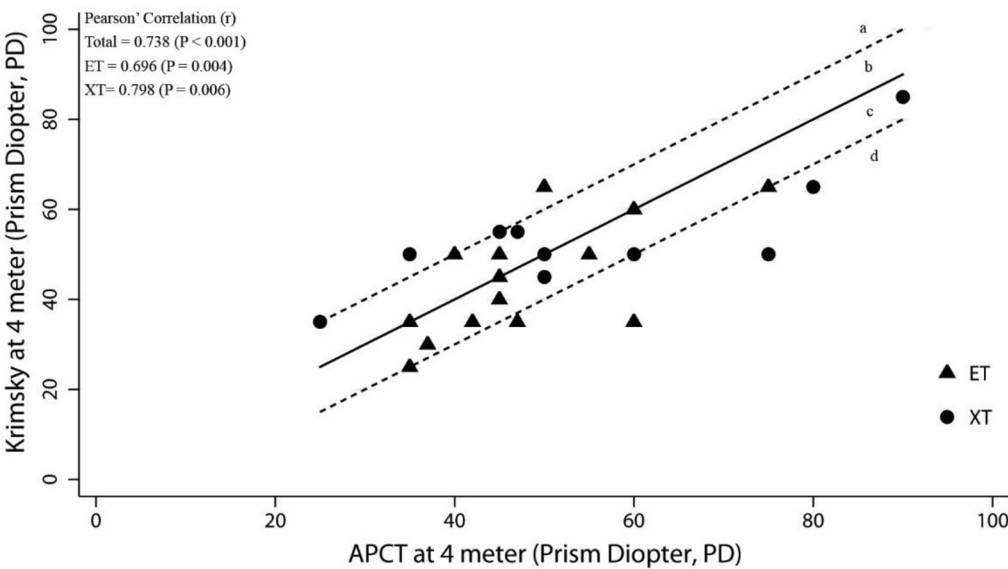

**Fig 6. The relationship between angle of deviation from the Krimsky test with APCT at D.** The solid line is the line of equality and the zone between dotted lines represents range ±10 PD of the APCT values. The acceptable area were b and c. The unacceptable area were a and d. (ET = esotropia; XT = exotropia).

This study compared the degree of angle of deviation from APCT and Photo-Hirschberg tests, with values of the Hirschberg coefficients from photographs and using specific software. This study also compared the degree of angle of deviation from APCT and Krimsky tests. Previous studies modified the Hirschberg test to improve the accuracy for measuring angle of deviation. The Photo-Hirschberg is one of such modifications, with moderately strong correlation with APCT for near and distance fixations (r = 0.415, 0.620). Half of the subjects had values within the acceptance range, but the correlation was lower compared with other studies. Yang [18] studied the efficacy of the 3D strabismus photo analyzer, computerized software for measuring angle of deviation from photographs. The results showed moderately strong correlation between an APCT and 3D strabismus photo analyzer (r = 0.772). Hasebe [12] studied an automated strabismus analysis from corneal light reflex using a video refractor. The results showed strong correlation(r = 0.956) between Hirschberg and APCT. This study defined an acceptance with discrepancy of deviation ±13.7 PD that differed from our study. Yoo [19], using infrared photographs, with a specific filter, measured the angle of deviation in small angle ET, and reported strong correlation(r = 0.815) with both APCT and Krimsky tests.

Sousa de Almeida [20], using a computer-aided methodology, based on detection of strabismus from a video, showed that the 95% limits of agreement were ±13.7 PD. The data from photographs may vary, because of intermittent XT patients, accommodative ET or error of dimension-distance [20, 21]. In intermittent XT patients, the value of deviation angle was less than that using APCT. We assumed that patients still had fusion while being photographed. Now, we suggest to improve the method of photographing to reduce accommodation, such as the use of an infrared filter.

The correlation between the Krimsky test and the APCT, for near fixation and distance fixation, were 0.773 and 0.738, respectively; indicating moderately strong correlation. The difference of the angle of deviation was within the acceptable range in three quarters of the subjects. Previous studies have reported similar correlation. Yang [18] showed strong correlation between APCT and Krimsky (r = 0.809), while Joo [22] showed moderately strong correlation

between APCT and Krimsky in ET and XT(r = 0.738, 0.651) and very strong correlation between APCT and Distance Krimsky tests in ET and XT(r = 0.981, 0.919).

The angle of deviation measured by the Photo-Hirschberg test, Krimsky test, and APCT showed positive correlation, suggesting that the Photo-Hirschberg method can be used as an alternative to APCT; especially in uncooperative patients. Additionally, it can be used as a follow up option for measuring the angle of deviation in strabismus subjects.

Limitations of this study, the number of study patients was quite small. However, the findings of this study can serve as a preliminary foundation for further study.

## Conclusion

The reliability of Krimsky test was better than Photo-Hirschberg test for measuring an angle of deviation.

## Supporting information

**S1 File. EC 56-167-02-4-2.**
(PDF)

**S2 File. Protocol EC 56-167-02-4-2.**
(PDF)

**S3 File. Translate protocol EC 56-167-02-4-2.**
(PDF)

**S4 File. EC 57-0117-02-7.**
(PDF)

**S5 File. Protocol EC 57-0117-02-7.**
(PDF)

**S6 File. Translate protocol EC 57-0117-02-7.**
(PDF)

**S7 File. Trend statement-TREND-check list.**
(PDF)

**S8 File. Raw data reliability.**
(PDF)

**S1 Table. Summary results of correlation between angle of deviation from the Photo-Hirschberg test with the APCT.**
(DOCX)

**S2 Table. Summary results of correlation between angle of deviation from the Krimsky with the APCT.**
(DOCX)

## Acknowledgments

We would like to thank the many individuals; including, Dr. Alan Geater for suggestions, statistical analysis and Ms. Parichat Damthongsuk for assistance in data collection and statistical analysis Additionally, thank you to Mr Andrew Tait, from the International Affairs department, for assistance in the editing of the English.

## Author Contributions

**Conceptualization:** S. Tengtrisorn.

**Data curation:** A. Tungsattayathitthan, S. Na Phatthalung, S. Bhurachokviwat, S. Chouyjan.

**Investigation:** S. Tengtrisorn.

**Methodology:** S. Tengtrisorn.

**Project administration:** S. Tengtrisorn.

**Writing – original draft:** S. Tengtrisorn.

**Writing – review & editing:** P. Singha, N. Rattanalert.

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
