## [Decision Letter · Decision Letter 0]

5 Jul 2021

PONE-D-21-11420

The reliability of the angle of deviation measurement from Photo-Hirschberg tests and Krimsky tests

PLOS ONE

Dear Dr. tengtrisorn,

Thank you for submitting your manuscript to PLOS ONE. After careful consideration, we feel that it has merit but does not fully meet PLOS ONE’s publication criteria as it currently stands. Therefore, we invite you to submit a revised version of the manuscript that addresses the points raised during the review process.

The idea of the manuscript is interesting However, the manuscript needs some justification for the exclusion of  some patients, and explanation of how patients were intermittent exotropia were photpgraphed..

We look forward to receiving your revised manuscript.

Kind regards,

Ahmed Awadein, MD, Ph.D, FRCS

Academic Editor

PLOS ONE

Journal Requirements:

2. Thank you for submitting your clinical trial to PLOS ONE and for providing the name of the registry and the registration number. The information in the registry entry suggests that your trial *TCTR20141201001* was registered after patient recruitment began. PLOS ONE strongly encourages authors to register all trials before recruiting the first participant in a study.

1) your reasons for your delay in registering this study (after enrolment of participants started);

2) confirmation that all related trials are registered by stating: “The authors confirm that all ongoing and related trials for this drug/intervention are registered”.

3. Please include your tables as part of your main manuscript and remove the individual files. Please note that supplementary tables should remain uploaded as separate "supporting information" files.

"This research was partially supported by a grant from the Faculty of Medicine, Prince of Songkla University."

"No"

"No"

7. We note that Figure 2 includes an image of a patient / participant in the study. 

Reviewers' comments:

Reviewer's Responses to Questions

**Comments to the Author**

1. Is the manuscript technically sound, and do the data support the conclusions?

Reviewer #1: Yes

Reviewer #2: Partly

2. Has the statistical analysis been performed appropriately and rigorously? 

Reviewer #1: Yes

Reviewer #2: No

3. Have the authors made all data underlying the findings in their manuscript fully available?

Reviewer #1: Yes

Reviewer #2: Yes

4. Is the manuscript presented in an intelligible fashion and written in standard English?

Reviewer #1: Yes

Reviewer #2: Yes

5. Review Comments to the Author

Reviewer #1: This study addresses an important issue and worked on a technique that may be useful in areas with no strabismus specialist. Although the idea is good yet the study is limited by the small number of cases and major exclusion criteria. The manuscript needs a lot of work to be suitable for publication. Below are the details.

Line 58: “Many studies have tried… “ What are those studies? No references mentioned.

Lines 55 to 82: This part should be moved to the discussion. The details of previous investigators’ approach can be included briefly in introduction only if needed to explain the purpose of the study. This section should be reduced in introduction and moved mostly to discussion.

Line 89: “Four subjects in the XT group were intermittent XT”: The authors described the diagnosis of four subjects out of the total cohort only. It is unclear why this diagnosis was singled out specifically here. Why not mention the breakdown of the diagnosis of all the test subjects?

Line 87-94: Inclusion criteria were mentioned, then diagnosis of 4 subjects, then the consent, then back to exclusion criteria. This order is not usual. I recommend keeping the exclusion criteria right after the inclusion criteria.

Lines 91-94: Why was accommodative ET excluded? Why were cases with amblyopia or pervious surgery excluded? These subgroups are very common diagnoses, especially in the difficult cases that will need to be sent to an expert for opinion. By such exclusions authors are suggesting that the test is no useful on these subgroups and hence not useful for a big chunk of the cases in the strabismus clinic. Cause of such exclusions should be discussed.

Lines 94-96: Revise sentence structure and avoid repetition (subjects received eye examination)

Line 96: I was expecting detailed description of how to Photo-Hirschberg test was done. This is mentioned later lines 108 onwards. I think this should be moved to be in the right place. It is confusing to have to go back and forth in the manuscript. As such it lacks the smoothness that readers expect.

Lines 99-100: “Time span..” It is not clear what the authors mean by time span. Time span between what and what? And why is that? This “time span” needs detailed description.

Lines 101 -106: The consent is repeated again here. It is mentioned earlier in line 90! I would recommend starting the methodology by stating the approval by the IRB and the consent and then proceeding with the rest of the methodology. Line 105: “and showed.. “ till the end should be deleted. At this point of reading the study didn’t show anything yet. We are still in the methods.

Line 117: “limbuson”: seems a spelling mistake

Line 118: “When the right eye is fixated”: I think the authors mean when the right eye “is fixing”. Please correct this throughout the text.

Line 108: How did the photographer deal with cases of intermittent exotropia to show the deviation?

Lines 170-187: All this data should be in detailed in the table and deleted from the text. It is very confusing to read or compare these number in paragraph format. A table only without text is enough to show results like this.

Lines 189-198: the first paragraph of the discussion is almost all a repetition from the introduction.

Lines 213-215: The angles of the cases should be in the results section. The analysis as to why this might have happened goes into the discussion section.

Lines 217: Do authors want to implicate that their method is not useful in intermittent exotropia cases? Need clarifications as to what is the authors conclusions about the use in intermittent exotropia cases.

Line 234: Conclusion should be in relation to the title and objective of the study. The study is testing “the reliability of Photo-Hirschberg and Krimsky”. The objective as per the abstract “to compare photo-H, Krimsky and APCT”. therefore this should be mentioned in the conclusion. This sentence “ Photo-Hirschberg test could be used to examine and monitor” is not informative. Any test could be used to measure and monitor! Same for saying “Krimsky can be used in uncooperative patients”: this is again basic knowledge and not the conclusion of this study.

Reviewer #2: The assessment of measurement reliability should not be done by Pearson's correlation but use the Intrasclass Correlation (ICC). Please refer to the recent comments on measurement reliability and its importance for individual differences research (https://www.nature.com/articles/s41562-019-0655-x). In https://www.sciencedirect.com/science/article/pii/S2095927318305784, the anatomy of ICC is introduced. Please update all the reliability analyses in terms of the basic requirement by the two references abovementioned.

6. PLOS authors have the option to publish the peer review history of their article (what does this mean?). If published, this will include your full peer review and any attached files.

Reviewer #1: **Yes: **Amr ElKamshoushy

Reviewer #2: No

---

## [Author Response · Author response to Decision Letter 0]

8 Aug 2021

The idea of the manuscript is interesting However, the manuscript needs some justification for the exclusion of some patients, and explanation of how patients were intermittent exotropia were photpgraphed..

ANS: We need to set the research protocol in real world, so intermittent exotropia was not in the exclusion criteria. After we included all subjects and analyzed, we try to explain for someone who needs to use the data in clinical practice. After your recommend we reanalyzed after excluded intermittent strabismus cases.

ANS: I changed my financial disclosure and include in cover letter.

We look forward to receiving your revised manuscript.

Kind regards,

Ahmed Awadein, MD, Ph.D, FRCS

Academic Editor

PLOS ONE

Journal Requirements:

2. Thank you for submitting your clinical trial to PLOS ONE and for providing the name of the registry and the registration number. The information in the registry entry suggests that your trial *TCTR20141201001* was registered after patient recruitment began. PLOS ONE strongly encourages authors to register all trials before recruiting the first participant in a study.

1) your reasons for your delay in registering this study (after enrolment of participants started); 

ANS: The ethics committee was approved the research protocol then we made registration of the protocol to TCTR before we started to recruit subjects. However, we did not check for completion of registration.

2) confirmation that all related trials are registered by stating: “The authors confirm that all ongoing and related trials for this drug/intervention are registered”.

 ANS: We confirmed that all related trials are registered by stating: “The authors confirm that all ongoing and related trials for this drug/intervention are registered”. 

3. Please include your tables as part of your main manuscript and remove the individual files. Please note that supplementary tables should remain uploaded as separate "supporting information" files.

ANS: Changed

"This research was partially supported by a grant from the Faculty of Medicine, Prince of Songkla University."- 

ANS: I rewrite it from the manuscript.

"No"

ANS: I include the statement “There was no additional external funding received for this study.” in my updated Funding Statement and put it in my cover letter

"No"

ANS: On behalf of all authors the corresponding author declares that no competing interests exist and put it in cover letter.

ANS: no other supporting information files

7. We note that Figure 2 includes an image of a patient / participant in the study. 

ANS: I changed the Figure 2 and no longer a photograph. 

Reviewers' comments:

Reviewer's Responses to Questions

Comments to the Author

1. Is the manuscript technically sound, and do the data support the conclusions?

Reviewer #1: Yes

Reviewer #2: Partly

2. Has the statistical analysis been performed appropriately and rigorously?

Reviewer #1: Yes

Reviewer #2: No 

ANS: We reanalyze the manuscript.________________________________________

3. Have the authors made all data underlying the findings in their manuscript fully available?

Reviewer #1: Yes

Reviewer #2: Yes

4. Is the manuscript presented in an intelligible fashion and written in standard English?

Reviewer #1: Yes

Reviewer #2: Yes

5. Review Comments to the Author

Reviewer #1: This study addresses an important issue and worked on a technique that may be useful in areas with no strabismus specialist. Although the idea is good yet the study is limited by the small number of cases and major exclusion criteria. The manuscript needs a lot of work to be suitable for publication. Below are the details.

Line 58: “Many studies have tried… “ What are those studies? No references mentioned.

Ans: Changed

Lines 55 to 82: This part should be moved to the discussion. The details of previous investigators’ approach can be included briefly in introduction only if needed to explain the purpose of the study. This section should be reduced in introduction and moved mostly to discussion.

ANS: Changed

Line 89: “Four subjects in the XT group were intermittent XT”: The authors described the diagnosis of four subjects out of the total cohort only. It is unclear why this diagnosis was singled out specifically here. Why not mention the breakdown of the diagnosis of all the test subjects? – 

ANS: We reanalyze and rewrite the manuscript after excluded intermittent strabismus cases. 

Line 87-94: Inclusion criteria were mentioned, then diagnosis of 4 subjects, then the consent, then back to exclusion criteria. This order is not usual. I recommend keeping the exclusion criteria right after the inclusion criteria. 

ANS: changed

Lines 91-94: Why was accommodative ET excluded? Why were cases with amblyopia or pervious surgery excluded? These subgroups are very common diagnoses, especially in the difficult cases that will need to be sent to an expert for opinion. By such exclusions authors are suggesting that the test is no useful on these subgroups and hence not useful for a big chunk of the cases in the strabismus clinic. Cause of such exclusions should be discussed.

ANS: Accmmodation ET showed variation of angle and depend on accommodation power and glasses correction so it may confound the interpretation.

Amblyopia case will have some problem for fixation when we measure angle of deviation 

from alternate prism cover test (APCT). 

Previous surgery may disturb the action of the eye muscle. 

Lines 94-96: Revise sentence structure and avoid repetition (subjects received eye examination) 

ANS: Changed

Line 96: I was expecting detailed description of how to Photo-Hirschberg test was done. This is mentioned later lines 108 onwards. I think this should be moved to be in the right place. It is confusing to have to go back and forth in the manuscript. As such it lacks the smoothness that readers expect.

ANS: Changed

Lines 99-100: “Time span..” It is not clear what the authors mean by time span. Time span between what and what? And why is that? This “time span” needs detailed description.

ANS: Changed

Lines 101 -106: The consent is repeated again here. It is mentioned earlier in line 90! I would recommend starting the methodology by stating the approval by the IRB and the consent and then proceeding with the rest of the methodology. Line 105: “and showed.. “ till the end should be deleted. At this point of reading the study didn’t show anything yet. We are still in the methods.

ANS: Changed

Line 117: “limbuson”: seems a spelling mistake

ANS: Changed

Line 118: “When the right eye is fixated”: I think the authors mean when the right eye “is fixing”. Please correct this throughout the text.

ANS: Changed

Line 108: How did the photographer deal with cases of intermittent exotropia to show the deviation?

ANS: We reanalyzed and rewrite the manuscript after excluded intermittent strabismus cases. We suggest to improve the method with using an infrared filter.

Lines 170-187: All this data should be in detailed in the table and deleted from the text. It is very confusing to read or compare these number in paragraph format. A table only without text is enough to show results like this.

ANS: Changed

Lines 189-198: the first paragraph of the discussion is almost all a repetition from the introduction.

ANS: Changed

Lines 213-215: The angles of the cases should be in the results section. The analysis as to why this might have happened goes into the discussion section.

ANS: Changed

Lines 217: Do authors want to implicate that their method is not useful in intermittent exotropia cases? Need clarifications as to what is the authors conclusions about the use in intermittent exotropia cases.

ANS: Changed

Line 234: Conclusion should be in relation to the title and objective of the study. The study is testing “the reliability of Photo-Hirschberg and Krimsky”. The objective as per the abstract “to compare photo-H, Krimsky and APCT”. therefore this should be mentioned in the conclusion. This sentence “ Photo-Hirschberg test could be used to examine and monitor” is not informative. Any test could be used to measure and monitor! Same for saying “Krimsky can be used in uncooperative patients”: this is again basic knowledge and not the conclusion of this study.

ANS: We changed conclusion. 

Reviewer #2: The assessment of measurement reliability should not be done by Pearson's correlation but use the Intrasclass Correlation (ICC). Please refer to the recent comments on measurement reliability and its importance for individual differences research (https://www.nature.com/articles/s41562-019-0655-x). In https://www.sciencedirect.com/science/article/pii/S2095927318305784, the anatomy of ICC is introduced. Please update all the reliability analyses in terms of the basic requirement by the two references abovementioned.

ANS: the reliability done by number of subjected within acceptable ranges ( within +/-10 PD of the APCT values). I study ICC from previous references but, I am not quite sure that its fit for our paper. 

6. PLOS authors have the option to publish the peer review history of their article (what does this mean?). If published, this will include your full peer review and any attached files.

---

## [Editor Report · Decision Letter 1]

5 Oct 2021

The reliability of the angle of deviation measurement from Photo-Hirschberg tests and Krimsky tests

PONE-D-21-11420R1

Dear Dr. tengtrisorn,

We’re pleased to inform you that your manuscript has been judged scientifically suitable for publication and will be formally accepted for publication once it meets all outstanding technical requirements.

Kind regards,

Ahmed Awadein, MD, Ph.D, FRCS

Academic Editor

PLOS ONE
---

## [Editor Report · Acceptance letter]

19 Nov 2021

PONE-D-21-11420R1 

The reliability of the angle of deviation measurement from the Photo-Hirschberg tests and Krimsky tests 

Dear Dr. Tengtrisorn:

I'm pleased to inform you that your manuscript has been deemed suitable for publication in PLOS ONE. Congratulations! Your manuscript is now with our production department. 

Kind regards, 

on behalf of

Dr. Ahmed Awadein 

Academic Editor

PLOS ONE